# Spoken Named Entity Localization as a Dense Prediction task: End-to-end Frame-Wise Entity Detection

## Abstract

Precise temporal localization of named entities in speech is crucial for privacy-preserving audio processing. However, prevailing cascaded pipelines propagate transcription errors and end-to-end models lack the temporal granularity required for reliable frame-level detection. To address these limitations, we introduce DEnSNEL (Dense End-to-end Spoken Named Entity Localizer), the first end-to-end model to perform direct, frame-level spoken named entity localization, without intermediate character or text representations. We reformulate the task as a dense, frame-wise binary classification ("entity" vs. "non-entity"), employing a lightweight encoder-classifier architecture. To improve boundary delineation, DEnSNEL incorporates a learnable complex filter bank to capture phonetic information, and employs a boundary-focused loss that explicitly optimizes span precision. On the SLUE Phase 2 benchmark, DEnSNEL outperforms state-of-the-art methods in frame-level spoken named entity localization, while requiring substantially fewer parameters. With its lightweight architecture and precise frame-level entity detection, DEnSNEL offers a practical and efficient solution for real-world privacy-sensitive speech applications. Our code and models will be released publicly.

## 1 Introduction

In recent years, large-scale multi-modal large language models (MLLMs) have begun to accept raw speech as input (e.g. voice-enabled LLMs), raising significant privacy concerns due to the potential exposure of sensitive information within speech inputs (Vecino et al., 2025). A key solution to mitigate this risk involves masking personally identifiable information (PII) directly within the speech signal prior to any higher-level processing. Another application is in the medical domain, where Cohn et al. (2019) introduce audio de-identification for clinical recordings, de-identifying doctor-patient audio conversations by redacting protected health information (PHI) in the waveform to preserve patient confidentiality. Therefore, the ability to precisely localize named entities within speech signals is crucial for the above applications. However, although text-based named entity recognition (NER) (Devlin et al., 2019b; Peters et al., 2018; Lample et al., 2016; Li et al., 2020) is well-established, spoken NER, particularly with accurate timestamp detection, remains underexplored. Among the relatively few studies on spoken NER, most focus on recognizing which entity types (person - PER, organization - ORG, etc.) appear in an automatic speech recognition (ASR) transcript (Shon et al., 2022; Ghannay et al., 2018; Yadav et al., 2020; Ayache et al., 2024), rather than pinpointing their precise temporal locations within the speech signal.

Traditional approaches to NER timestamp detection rely on pipeline systems (Cohn et al., 2019; Baril et al., 2022), which can lead to ASR error propagation to subsequent NER stages and optimization remaining suboptimal due to independent training objectives. Recent end-to-end approaches (Shon et al., 2023; Arora et al., 2024), formally define this task as named entity localization (NEL), predicting the start and end timestamps of named entities in a speech utterance. They treat it as a general spoken language understanding (SLU) task. Such SLU-focused methods trained with connectionist temporal classification (CTC) (Graves et al., 2006) rely on a generic CTC recognizer to produce character-level sequences augmented with entity-type boundary tokens (e.g., "#"/"]" for ORG start/end) (Shon et al., 2023) and do not train a dedicated NEL model. However, CTC is

Figure 1: Overview of DEnSNEL for spoken named entity localization and masking. A time-aligned binary mask is predicted from the input audio and used to redact entity segments for privacy-preserving speech processing.

not explicitly trained for the precise temporal alignment which is crucial for effective data masking (Shon et al., 2023), and broader SLU objectives may introduce unnecessary complexity for the specific application of privacy preservation. Since these approaches classify entities into types and compute character-level representations, they perform unnecessary work for privacy masking, which only requires knowing where entities occur. Thus, there is a clear need for a lightweight, low-latency, end-to-end approach that can run directly on-device to detect and localize entity spans in audio, enabling frame-level redaction without involving text.

In this work, to the best of our knowledge, we present the first end-to-end approach to perform direct, frame-level spoken named entity localization, without intermediate character or text representations. We address this task through a paradigm shift in spoken NEL formulation. Rather than treating entity localization as a byproduct of SLU, we frame it as a token-level dense prediction task. A single lightweight encoder-classifier network processes audio and produces a frame-wise binary label: "entity" vs. "non-entity," without ever generating text. This focus on binary detection simplifies the task for privacy masking (see figure 1).

To enhance the sensitivity to subtle speech cues, such as sudden spectral changes that often indicate entity boundaries, we incorporate an explicit phonetic feature extraction function. We achieve this by implementing a small learnable complex filter bank (Ravanelli & Bengio, 2018; Zeghidour et al., 2018). This filter bank learns Gaussian-windowed sinusoidal filters on the mel scale, creating time-frequency representations that highlight phonetic transitions. While the encoder provides contextual representations capturing higher-level semantic information, the learnable complex filter bank extracts detailed time-domain transitions. These transitions help to precisely define the boundaries of entities. This joint optimization improves the model's accuracy in pinpointing the exact start and end timestamps of each spoken entity.

Building on this formulation, we view spoken NEL as a one-dimensional dense prediction task over the temporal axis. Drawing inspiration from 2D image segmentation and object detection, we train DEnSNEL with a composite loss that blends frame-wise binary cross-entropy (BCE) and an overlap-based temporal intersection-over-union (IoU) term. The BCE component enforces correct per-frame semantic classification, while the temporal IoU term explicitly rewards high overlap between predicted and ground-truth spans, thereby sharpening start and end boundary alignment.

We summarize our key contributions as follows:

- Reformulate NEL as a dense, frame-wise binary prediction task, detecting "entity" vs. "non-entity" directly in the audio stream, without intermediate character or text representations.
- Phonetic-enhanced feature integration for precise entity boundary detection.
- Optimized training with temporal boundary alignment loss (TBAL), combining binary cross-entropy and temporal intersection-over-union for improved semantic accuracy and precise alignment of entity boundaries.

## 2 RELATED WORK

### 2.1 SPOKEN NAMED ENTITY RECOGNITION

Named Entity Recognition is a fundamental task in natural language processing (NLP) that involves identifying and classifying entities such as person, organization, and location within textual data.

Early work focuses on rule-based or statistical models, while recent approaches significantly advance NER performance through the advent of transformer-based models (Devlin et al., 2019a; Liu et al., 2019). In contrast, NER from speech is a relatively less explored domain. Traditional approaches to this task rely on a sequential pipeline, where automatic speech recognition (ASR) systems first generate text transcripts, followed by text-based NER models to identify entities (Pasad et al., 2022). Although straightforward, this decoupled approach suffers from ASR error propagation, degraded temporal alignment, and suboptimal training due to independent training objectives (Yadav et al., 2020; Ghannay et al., 2018). Recent end-to-end approaches for spoken NER (Ghannay et al., 2018; Yadav et al., 2020; Shon et al., 2022; Ayache et al., 2024; Meeus et al., 2024; Yu et al., 2025b; Arora et al., 2022) address these limitations through joint optimization of ASR and NER within a unified framework, achieving comparable performance to pipeline approaches (Yu et al., 2025a). However, these approaches primarily focus on entity classification rather than precise temporal localization of named entities.

## 2.2 Spoken named entity localization

This task involves detecting named entities in speech and determining their start and end timestamps directly from the audio signal. Audio de-ID (Cohn et al., 2019) first defines this as audio de-identification, detecting and redacting Personal Health Identifiers (PHI) in clinical conversations using an ASR-NER pipeline followed by text-to-audio alignment. Similarly, Baril et al. (2022) extends this approach to French, combining a forced aligner with text-based NER to anonymize audio by silencing detected entity segments. Shon et al. (2023) introduce SLUE Phase-2, which defines Named Entity Localization (NEL) as predicting the start and end timestamps of named entities, extending SLUE's (Shon et al., 2022) NER task to frame-level span prediction. Rather than training a dedicated NEL model, their end-to-end baseline re-purposes a CTC-based ASR model by extending its output vocabulary with special entity boundary tokens. By following the same method as SLUE Phase-2, (Arora et al., 2024) evaluate a variety of supervised and self-supervised speech foundation models (SFMs) on the SLUE tasks, including NEL.

None of these models are explicitly trained with a temporal-localization objective; all rely on CTC-derived timestamps and post-hoc heuristics, underscoring the need for methods that directly optimize frame-wise entity detection.

## 2.3 Phonetic feature extraction

While prior work shows the utility of fixed or learnable filterbanks for phonetic analysis, few methods explicitly align phonetic cues to downstream temporal tasks. Ravanelli & Bengio (2018) first demonstrates that SincNet-style band-limited filters, initialized on the mel scale, can be learned end-to-end to capture speaker-specific and phonetic characteristics. Zeghidour et al. (2018) extends this to phone recognition by learning time-domain filterbanks with Gaussian envelopes, improving robustness under variable acoustic conditions. Mallat's wavelet framework (Mallat, 1999) and scattering transforms (Oyallon & Mallat, 2015) further show that Gabor-like filters emphasize formants and transient events critical for precise boundary detection.

These studies collectively suggest that aligning learnable, band-limited filters to frame-wise phenomena can sharpen temporal transitions by jointly optimizing complex Gabor filters with a boundary-focused loss.

## 3 Methodology

DEnSNEL departs from text-based approaches by defining each mel-spectrogram frame as an individual audio token, preserving temporal alignment and enabling precise, frame-wise localization of named entities directly from the waveform. In this section, we present our methodology for extracting contextual and phonetic features from speech, followed by frame-wise entity detection and temporal alignment through temporal boundary alignment loss (TBAL). An overview of our DEnSNEL architecture is presented in Figure 2.

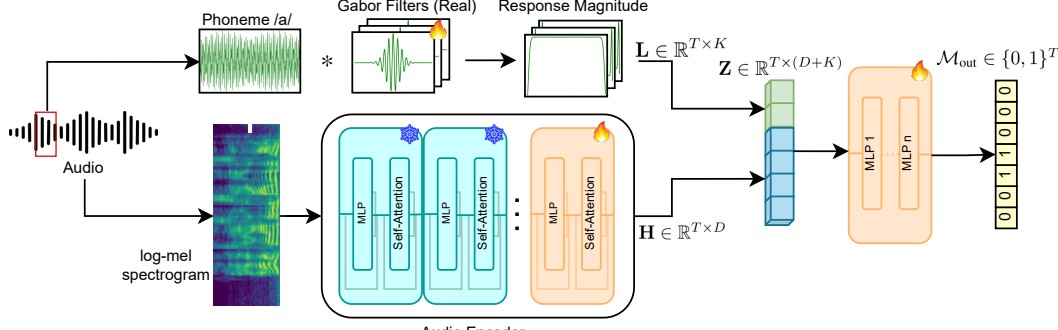

Figure 2: Overview of the DEnSNEL architecture: Phonetic and contextual features are extracted from raw audio and mel-spectrograms, fused, and classified to predict frame-wise entity masks.

## 3.1 CONTEXUAL FEATURE EXTRACTION

Given an input speech signal, we first compute its mel-spectrogram representation, denoted as $\mathbf{M} \in \mathbb{R}^{T \times F}$, where $T$ represents the number of frames and $F$ is the number of mel frequency bins. Each frame, corresponding to a $\delta t$ time window, acts as an audio token, ensuring fine-grained temporal alignment between input and output.

We leverage a pretrained audio encoder $\mathcal{E}$ as the backbone for feature extraction. The encoder processes the mel-spectrogram sequence and outputs contextual embeddings;

$$\mathbf{H} = \mathcal{E}(\mathbf{M})$$

where $\mathbf{H} \in \mathbb{R}^{T \times D}$ and $D$ is the configurable embedding size. Preserving the temporal resolution, the contextual embeddings $\mathbf{H}$ serve as an input to the classification head.

## 3.2 PHONETIC FEATURE EXTRACTION

To complement the spectral representations learned by the audio encoder $\mathcal{E}$, we introduce a *Learnable Complex Filterbank* that directly operates on raw waveform inputs to capture fine-grained phonetic cues. This module is inspired by Gabor-based and SincNet-style filterbanks (Ravanelli & Bengio, 2018; Zeghidour et al., 2018), but extends them with fully learnable center frequencies and Q-factors initialized on the mel scale.

**Filter initialization and parameterization**   We first define $K$ band-limited filters whose center frequencies $f_{c,k}$ are uniformly spaced to cover perceptual frequency bands (Stevens et al., 1937). Rather than learning $f_{c,k}$ and $b_k$ directly (where $b_k$ being the bandwidth initialized with 3 dB), we re-parameterize each filter by its log-center frequency $\log f_{c,k}$ and Q-factor $Q_k = f_{c,k}/b_k$. This log-domain representation guarantees strictly positive frequencies and enforces a lower bound on bandwidths, which mitigates filter collapse during training (Ravanelli & Bengio, 2018; Zeghidour et al., 2018). Each complex Gabor filter $a_k(t)$ as a Gaussian-windowed sinusoid:

$$a_k(t) = \exp\left(-\frac{t^2}{2\sigma_k^2}\right) \exp\left(j\,2\pi f_{c,k}t\right)$$

$$a_k(t) = w_k(t)\left[\cos(2\pi f_{c,k}t) + j\,\sin(2\pi f_{c,k}t)\right]$$

where $\sigma_k = \frac{1}{\pi\,b_k}$ is the Gaussian bandwidth and $w_k(t) = \exp(-t^2/(2\sigma_k^2))$ (Mallat, 1999; Oyallon & Mallat, 2015).

**Convolution and feature extraction.**   Given an input waveform segment $x(t)$, the real and imaginary parts of the filter responses are obtained by 1D convolutions ($*$):

$$y_{\text{re},k}(t) = (x * \Re\{a_k\})(t)$$

$$y_{\text{im},k}(t) = (x * \Im\{a_k\})(t)$$

We then compute the instantaneous magnitude (i.e. spectral envelope) to obtain a phase-invariant representation that emphasizes the energy distribution across frequency bands, accentuating formant transitions and transient phonetic cues crucial for precise boundary detection.

A non-linear activation (GELU (Hendrycks & Gimpel, 2016)) and layer normalization over the filter dimension follow to yield the final phonetic feature map $\mathbf{L} \in \mathbb{R}^{T \times K}$.

By directly learning band-limited Gabor filters, the learnable complex filterbank emphasizes formant structures and transient phonetic events—information that mel-spectrograms may blur, especially under variable acoustic conditions (Zeghidour et al., 2018).

Our learnable complex filterbank architecture extends previous learnable filterbank approaches by jointly learning both center frequencies and Q-factors of Gaussian-windowed sinusoidal filters initialized on the mel scale. Unlike prior works (Zeghidour et al., 2018; Ravanelli & Bengio, 2018) that primarily focus on either fixed or partially learnable filter parameters, our method fully adapts the filter shapes to better capture fine-grained phonetic transitions and formant structures directly from raw waveforms. This leads to improved delineation of spoken entity boundaries and enhanced phonetic representation (see figure 5), which is particularly beneficial for frame-level spoken named entity localization tasks.

### 3.3 FRAME-WISE ENTITY DETECTION

We concatenate the contextual and phonetic streams to form an enriched feature representation; $\mathbf{Z} = [\, \mathbf{H} \, , \, \mathbf{L} \,] \in \mathbb{R}^{T \times (D+K)}$.

This joint feature representation is passed through a lightweight classification head $\mathcal{C}$, to estimate the probability of entity presence at each frame. The model thus leverages both global context and local phonetic detail to make robust, temporally precise predictions. $\hat{y}_t = \sigma(\mathcal{C}(z_t))$, $z_t \in \mathbf{Z}$. where $\hat{y}_t \in [0, 1]$ represents the predicted probability for the presence of an entity in the $t$-th token, and $\sigma$ denotes the sigmoid activation function.

The model outputs a sequence of predictions $\{\hat{y}_t\}_{t=1}^{T}$ for each audio frame. These predictions are converted into a binary mask $\mathcal{M}_{\text{out}} \in \{0, 1\}^T$ by thresholding each $\hat{y}_t$ by a predefined threshold $\tau$.

Rather than relying on implicit alignment, DEnSNEL preserves exact frame-to-waveform correspondence throughout its end-to-end design and further tightens this link via our temporal boundary alignment loss (TBAL) (see section 3.4). By combining contextual representations from a fine-tuned pretrained encoder with phonetic features from a learnable filterbank, and coupling them with a lightweight classification head, our approach achieves high temporal resolution without significant computational overhead.

### 3.4 TEMPORAL BOUNDARY ALIGNMENT LOSS (TBAL)

To achieve precise temporal localization in DEnSNEL, we propose a *temporal boundary alignment loss (TBAL)*. This loss explicitly penalizes discrepancies between the predicted and the ground-truth temporal boundaries of named entities, ensuring that the model's predictions are not only semantically correct but also temporally precise.

As described in Section 3.3, we derive the binary prediction mask $\mathcal{M}_{\text{out}}$ from the predicted probability sequence $\hat{\mathbf{y}} = \{\hat{y}_t\}_{t=1}^{T} \in [0, 1]^T$ by applying a threshold $\tau$. Similarly, the ground-truth binary mask $\mathbf{g} = \{g_t\}_{t=1}^{T} \in \{0, 1\}^T$ is defined such that $g_t = 1$ if an entity is present in frame $t$ and 0 otherwise.

We define the overlap between the predicted and ground-truth masks via the temporal intersection $I_t$ and union $U_t$:

$$I_{\text{t}} = \sum_{t=1}^{T} \hat{y}_t \cdot g_t$$

$$U_{\text{t}} = \sum_{t=1}^{T} \hat{y}_t + \sum_{t=1}^{T} g_t - \sum_{t=1}^{T} \hat{y}_t \cdot g_t$$

Table 1: Frame-wise NEL F1 scores on the test dataset of NEL task in SLUE Phase 2 benchmark Shon et al. (2023). We present the results for models used in SLUE Phase 2 Shon et al. (2023) and SLUE-PERB Arora et al. (2024) along with our DEnSNEL. We achieve state-of-the-art results even when using the 'small' variant of Whisper Radford et al. (2023) as the encoder.

| | Configuration | NEL Frame F1 ↑ |
|---|---|---|
| SLUE Phase 2 Shon et al. (2023) | pipeline-w2v2 | 65.2 |
| | E2E-w2v2 | 56.3 |
| | pipeline-nemo | 74.1 |
| SLUE-PERB Arora et al. (2024) | HuBERT (large) | 69.8 |
| | Wav2Vec2 (large) | 71.2 |
| | WavLM (large) | 72.6 |
| | Whisper (medium) | 71.8 |
| | OWSM (3.1) | 70.5 |
| | Pre-trained SLU | 54.4 |
| DEnSNEL (Ours) | Whisper (tiny) | 60.9 |
| | Whisper (base) | 69.3 |
| | WavLM (large) | 75.9 |
| | Whisper (small) | 76.7 |
| | Whisper (medium) | **78.4** |

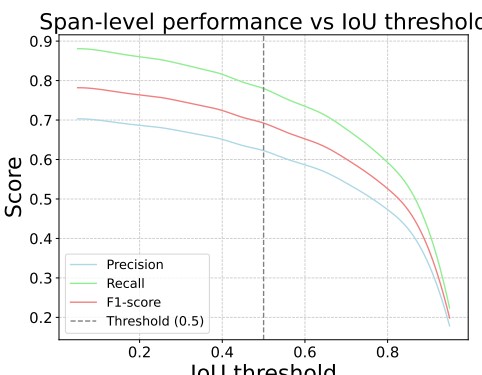

Figure 3: Span-level model performance across varying intersection-over-union thresholds (for the best-performing DEnSNEL (Whisper-small) configuration).

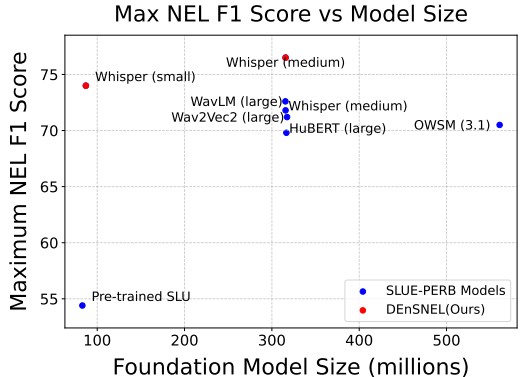

Figure 4: Relationship between foundational model size and NEL Frame F1 score.

The temporal intersection measures the number of true entity frames correctly predicted (i.e., true positives). By maximizing $I_t$, thereby minimizing false negatives, the model increases recall and ensures it captures as many actual entity frames as possible. The temporal union measures the total span covered by either prediction or ground truth, and penalizing large $U_t$ values discourages over-prediction (i.e., moderates false positives), by shrinking the combined span when predictions extend beyond true entity boundaries. We define the temporal intersection over union (IoU) loss:

$$\mathcal{L}_{\text{tIoU}} = 1 - \frac{I_t}{U_t}$$

$\mathcal{L}_{\text{tIoU}}$, enforces fine-grained alignment between predicted and ground-truth spans through overlap-based supervision, promoting accurate localization.

We combine $\mathcal{L}_{\text{tIoU}}$ with binary cross-entropy loss $\mathcal{L}_{\text{BCE}}$ in a weighted sum:

$$\mathcal{L}_{TBAL} = (1 - \beta)\,\mathcal{L}_{\text{BCE}} + \beta\,\mathcal{L}_{\text{tIoU}}$$

where $\beta \in [0, 1]$ is a hyperparameter that balances the two loss components. This jointly optimizes both semantic accuracy and temporal localization, ultimately leading to improved performance in spoken NEL.

# 4 EXPERIMENTS AND RESULTS

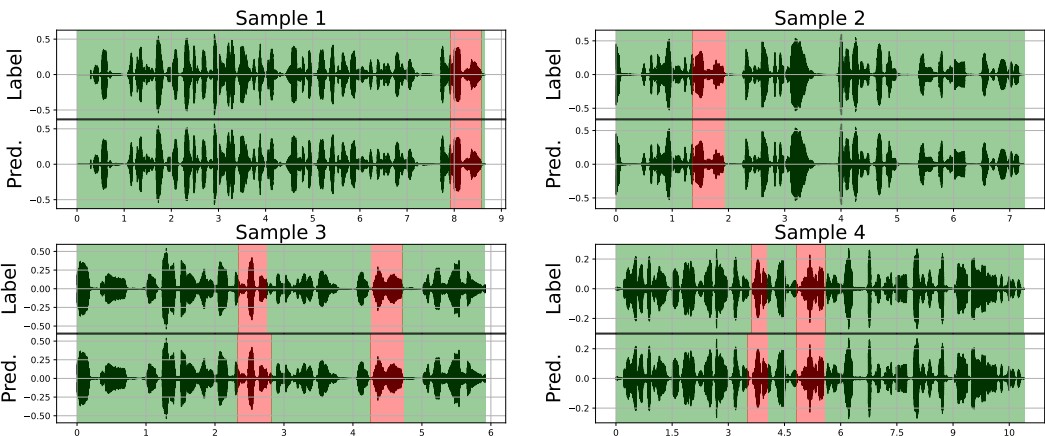

Figure 5: **Qualitative Evaluation:** Red regions indicate the presence of named entities in both ground truth (Label) and prediction (Pred), while green regions denote non-entity spans. We highlight how DEnSNEL aligns entity spans accurately, showcasing precise and consistent predictions.

## 4.1 THE DATASET

We evaluate our approach on the SLUE-VoxPopuli NEL test set from the SLUE Phase 2 benchmark (Shon et al., 2023). As no official training data is available, we construct our own training dataset by augmenting the SLUE-VoxPopuli NER training set (Shon et al., 2022) with word-level timestamps. We employ Montreal Forced Aligner (MFA) (McAuliffe et al., 2017) to obtain these word-level time stamps. (Further details are provided in Appendix A.3)

## 4.2 DENSNEL CONFIGURATION

We use the Whisper encoder (Radford et al., 2023) as the audio encoder in our DEnSNEL architecture. We evaluate our approach using the Whisper medium, small, base, and tiny variants, fine-tuning only a subset of final encoder layers. We maintain an approximately constant ratio of 1:6 between fine-tuned and total encoder layers across all variants.

The experimental setup for phonetic feature extraction involves implementing a custom learnable complex filterbank, drawing inspiration from Gabor and SincNet architectures (Zeghidour et al., 2018), which processes raw 16 kHz audio. This filterbank generates 32 feature channels using 251-sample kernels and a stride of 320, with learnable log-center frequencies and Q-factors initialized on the mel scale. The feature processing pipeline involves convolving the audio with the filterbank's real and imaginary components, computing the instantaneous magnitude, applying GELU activation, permuting dimensions, and finally, applying layer normalization.

We concatenate the contextual feature representations from the audio encoder with phonetic features derived from the learnable complex filter banks and pass them through a 3-layer multilayer perceptron (MLP) with a dropout rate of 0.1, where each hidden layer is followed by layer normalization and ReLU activation (see section 4.4).

**Implementation details:** We utilize an AdamW optimizer with a learning rate of $1 \times 10^{-5}$ and a weight decay of $1 \times 10^{-4}$. The model undergoes training for up to 50 epochs with early stopping (patience = 8) based on validation loss convergence. We set the batch size to 8 and execute all experiments on a NVIDIA RTX 4000 GPU. We used a fixed threshold of $\tau = 0.5$ for all the evaluations.

## 4.3 RESULTS

DEnSNEL achieves state-of-the-art performance on the SLUE Phase 2 benchmark (Shon et al., 2023) for spoken NEL. With the Whisper small encoder, it surpasses previous best performing Whisper medium-based models by +4.9 F1 and outperforms the strongest WavLM large-based model by +4.1 F1, while using 67% fewer parameters. Using the Whisper medium encoder, it achieves the highest overall performance, exceeding the best result reported in Arora et al. (2024) by +5.8 F1 score. We integrate the top performing WavLM (large) encoder from Arora et al. (2024) and observe a +3.3 F1 improvement in our architecture. Even with compact encoders like Whisper base and tiny, DEnSNEL demonstrates competitive performance, highlighting its efficiency and scalability across model sizes. The results are summarized in Table 1.

The inference process for a 5 to 15 second audio clip requires approximately 0.1 seconds when processed on a GPU (specifications provided in Section 4.2) and around 0.5 seconds on a CPU (More details in Appendix A.4). This low-latency inference makes DEnSNEL computationally efficient enough to be suitable for near real-time applications.

In privacy-sensitive speech applications, recall is prioritized over precision, since missing an entity poses greater risks than over-masking non-entities. Accordingly, DEnSNEL sustains substantially high recall than precision even at its optimal F1 (see Figure 3 and more details in Appendix A.2.1).

DEnSNEL maintains strong span-level performance across a wide range of IoU thresholds (Figure 3). We match predicted spans to ground-truth spans using a predefined threshold $\theta$ and evaluate localization quality with precision, recall, and F1-score. The model achieves high performance up to thresholds of 0.7–0.8, and performance declines at stricter overlaps as fewer predicted spans satisfy the higher IoU requirement. This reflects the trade-off between minor misalignment tolerance and strict boundary precision, showing that DEnSNEL delivers accurate, temporally precise predictions for applications prioritizing exact entity span localization.

In DEnSNEL, we adopt the audio encoder of a small-scale foundation model ($\sim$87M parameters) combined with a lightweight 3-layer MLP classification head, surpassing existing approaches, which mostly rely on large-scale foundation models such as Wav2Vec2 (large), HuBERT (large), and WavLM (large) combined with either a simple linear classifier or a complex encoder-decoder architecture. These results highlight the effectiveness of our task reformulation, phonetic feature integration, and temporal boundary alignment loss, along with the lightweight MLP head.

**Zero-shot performance on BLAB benchmark**

We further evaluate DEnSNEL on the long-form audio benchmark introduced in Ahia et al. (2025). Although their reported results use large MLLMs, these models are not directly comparable to our objective of on-device named entity detection and masking prior to transmitting audio to such systems. Moreover, the entity categories defined in their benchmark do not fully correspond to those used in our model. Notably, DEnSNEL achieves either substantially higher or comparable performance based on the evaluation setting. Additional details are provided in the Appendix A.1.

## 4.4 ABLATIVE STUDY

**Classification head** We assess the impact of different classification head architectures on NEL performance in Table 3. Our 3-layer MLP head achieves the highest Frame F1 score while using substantially fewer parameters compared to the convolutional and transformer-based counterparts. We attribute the MLP's superior performance to its simplicity and effective regularization, helps us prevent overfitting and achieve more precise boundary localization. In contrast, although the transformer and convolutional heads offer higher capacity, we hypothesize spatial smoothing may reduce localization accuracy.

**Loss function** We evaluate the impact of different loss functions on the frame-level NEL performance, in Table 2. Using temporal intersection over union (tIoU) loss alone is slightly better than using standard binary cross entropy (BCE) loss alone, indicating that explicitly supervising the temporal overlap between predicted and ground-truth spans enhances boundary alignment. However, the best performance is achieved by our proposed TBAL loss, which combines BCE and tIoU losses. This joint formulation guides the model to both classify entity frames accurately and localize entity boundaries precisely by balancing fine-grained frame-level supervision with global span-level align-

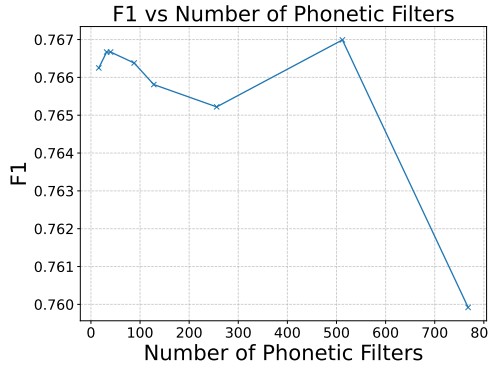

Figure 6: Effect of the number of trainable phonetic filters on NEL F1 score.

Table 2: Comparison of loss functions for DEnSNEL. Combining BCE and temporal IoU into the proposed TBAL loss ($\mathcal{L}_{\text{TBAL}}$) leads to the best performance, highlighting the importance of jointly modeling label alignment and segment-level overlap.

|  | Loss function | NEL Frame F1 |
|---|---|---|
| **(I)** | BCE loss ($\mathcal{L}_{\text{BCE}}$) | 75.1 |
| **(II)** | Temporal IoU loss ($\mathcal{L}_{\text{tIoU}}$) | 75.8 |
| **(III)** | TBAL loss ($\mathcal{L}_{\text{TBAL}}$) (**(I)** + **(II)**) | **76.7** |

Table 3: Effect of classification heads on NEL Frame F1. The 3-layer MLP head achieves the best performance while being significantly more parameter-efficient.

| Classification head | Params (M) | NEL Frame F1 |
|---|---|---|
| 5-Layer CNN w/ BN + ReLU | 6.01 | 76.1 |
| 2-Layer Transformer + MLP | 15.90 | 76.4 |
| 3-Layer MLP w/ Dropout + LN + ReLU | 0.48 | **76.7** |

ment. We attribute this improvement to TBAL's capacity to promote temporal consistency across frames, reducing fragmented or overly discrete span predictions.

**Number of phonetic filters** We analyze the impact of varying the number of trainable phonetic filters on the NEL F1 score in Figure 6. The performance peaks at 512 filters, indicating strong representational capacity. Notably, 32 filters achieve nearly equivalent performance, followed by 40 and 88, suggesting that even compact configurations can effectively capture key phonetic features. Beyond 256 filters, the diminishing gains, and the drop at 768, suggest that additional filters may introduce noise or redundancy. Larger filter banks also incur higher computational costs. Hence, we adopt 32 filters as the optimal trade-off between accuracy and efficiency.

## 5 CONCLUSION AND LIMITATIONS

In this work, we present DEnSNEL, the first text-independent, end-to-end frame-wise spoken named entity localizer for privacy-preserving applications. Operating directly on speech, it predicts dense frame-level binary labels using an audio encoder enhanced with a learnable complex filter bank and a boundary-aware training objective. On the SLUE-Phase 2 benchmark, DEnSNEL improves frame-level F1 by +4.1 over prior state-of-the-art methods with 67% fewer parameters, and demonstrates competitive zero-shot performance on the long-form BLAB benchmark (Ahia et al., 2025).

Our evaluation is limited by the scarcity of publicly available benchmarks with diverse acoustic and linguistic conditions. Both the SLUE benchmark and our training set rely on MFA-generated timestamps (McAuliffe et al., 2017), which may introduce alignment biases due to the lack of manual annotations. Additionally, very few prior works provide publicly available code or reproducible setups, making direct comparisons challenging. As discussed in Section 4.3, standard metrics such as balanced F1 may not fully capture practical utility for privacy-critical applications, where maximizing recall is often more important than precision.

Future work should explore more diverse datasets, manually annotated benchmarks, and recall-oriented evaluation metrics to better reflect real-world needs. Overall, DEnSNEL establishes a foundational step toward lightweight, privacy-preserving speech processing systems capable of accurate, frame-level named entity localization directly on audio.

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

# A APPENDIX

## A.1 FURTHER RESULTS

BLAB (Ahia et al., 2025) introduces a challenging long-form audio benchmark, reporting results using both open-source and proprietary audio LMs, including Gemini 2.0 Pro and GPT-4o. This differs from our motivation, which focuses on lightweight, on-device named entity detection and masking prior to sending audio to those large MLLMs. Moreover, BLAB focuses on long-form recordings, averaging 51 minutes in length, which was not part of our initial design goals; however, DEnSNEL can be evaluated under these conditions. To examine the performance of DEnSNEL under long-context conditions, we split the recordings into 30-second segments and processed them sequentially.

The entity categories in BLAB (Event, Location, NORP, Organization, Person, TV Show, Temporal, and Work of Art) do not fully correspond to those used in our approach. We therefore restrict our analysis to the 49 recordings in the "All Entities" category, which contains all nine entity types, while the remaining files consist of single-category recordings. Additionally, BLAB's dataset curation relies on WhisperX (Bain et al., 2023) for transcription and alignment, followed by Claude-3 (Anthropic, 2024) for text-based NER. Consequently, the accuracy of the ground-truth entity spans is potentially biased. Therefore, direct performance comparisons with BLAB results are not strictly fair.

Our evaluation does not correspond to BLAB-MINI, the short-audio subset of BLAB with audio segments under 30 seconds, which is not given in detail. Instead, we process long recordings in 30-second chunks, resulting in a hybrid evaluation setting. Despite these differences, DEnSNEL achieves either substantially higher or comparable performance relative to the much larger audio LMs reported on BLAB. We emphasize, however, that direct comparison to these large models falls outside the primary scope of our work, which focuses on on-device efficiency and preprocessing audio before it is sent to large-scale MLLMs, rather than competing with them.

| Dataset | Gemini 2.0 Flash | Gemini 2.0 Pro | Qwen 2.0 | Phi-4 | Ours |
|---------|------------------|----------------|----------|-------|------|
| BLAB MINI | 45.49 | 49.58 | 12.07 | 7.63 | 40.66 |
| BLAB | 2.97 | 2.14 | – | – | |

Table 4: Zero-shot frame-wise NEL F1 scores on the NEL task in BLAB benchmark (Ahia et al., 2025).

## A.2 DEnSNEL FROM AN APPLICATION PERSPECTIVE

DEnSNEL is a lightweight, end-to-end approach for frame-level spoken named entity localization. Its strong performance and efficient architecture enable its use in a broad range of applications, including privacy-preserving speech processing.

The on-device deployability of DEnSNEL is another critical factor, particularly in edge computing scenarios where data privacy and low latency are paramount. Lightweight models are essential for on-device processing due to the limited computational resources and battery life of mobile devices and embedded systems. By deploying DEnSNEL directly on the device, sensitive voice data can be processed locally without being transmitted to a remote server, thereby mitigating the risk of eavesdropping or data breaches. This is particularly important in applications such as voice-based authentication, where biometric data must be protected, and in healthcare applications, where patient confidentiality is crucial.

Moreover, DEnSNEL enables near real-time processing (approximately 0.1 seconds for a 10-second audio), which is essential for applications that require immediate responses. For example, in voice-activated personal assistants, a lightweight model can quickly identify and link named entities in spoken queries, providing users with timely and relevant information.

In summary, DEnSNEL plays a vital role in enhancing the security and privacy of voice-controlled systems and enabling on-device processing for a wide range of applications. The development of lightweight models are crucial for achieving near real-time performance and protecting sensitive voice data in edge computing environments. As voice-based interfaces become increasingly prevalent, the importance of DEnSNEL will continue to grow, driving further research and innovation in this field.

### A.2.1 PRIORITIZING RECALL IN PRIVACY-SENSITIVE SPEECH APPLICATIONS

In privacy-sensitive speech applications, missing an actual entity (a false negative) poses far greater risk than over-masking benign content (a false positive). This is because unmasked entities can lead to privacy breaches or regulatory violations, whereas masking non-entities merely affects readability. Consequently, maximizing recall, ensuring that nearly all true entity frames are detected is paramount, even at the expense of precision. Notably, DEnSNEL sustains substantially higher recall than precision, even at its optimal F1, reflecting a deliberate bias toward detecting entity spans, an essential property when missing an entity is far more costly than over-masking (see Figure 3).

### A.3 THE DATASET

We focus on the task of Named Entity Localization (NEL), a relatively underexplored area with limited access to suitable datasets. Among existing efforts, Cohn et al. (2019) introduce the SWFI benchmark, which includes annotated segments from the Switchboard (Godfrey et al., 1992) and Fisher (Cieri et al., 2004) conversational English corpora. However, these datasets are not freely available, limiting their utility for broader research. Baril et al. (2022) present an annotated French corpus with word boundaries. Shon et al. (2023) provide a publicly available English dataset as part of the SLUE benchmark, but only the test and validation splits are accessible; the training set remains unavailable. This scarcity of comprehensive, open datasets highlights the need for more accessible resources to advance research in NEL.

We evaluate our method on the SLUE-VoxPopuli NEL test set, released in Phase 2 of the SLUE benchmark (Shon et al., 2023). Since no official training set for NEL exists, we construct our own. The NEL task in Shon et al. (2023) extends the Named Entity Recognition (NER) setup from Shon et al. (2022), primarily by incorporating word-level timestamps. Based on this, we augment the SLUE-VoxPopuli NER training set (Shon et al., 2022) with word-level alignments obtained using the Montreal Forced Aligner (MFA) (McAuliffe et al., 2017), following the same alignment methodology used in Shon et al. (2023).

### A.3.1 ERROR ANALYSIS OF USING MFA FOR ALIGNMENT

We performed a simple manual check of 20 randomly sampled utterances in the training dataset. We compared the MFA-generated named entity timestamps with manually verified boundaries and observed the following:

- Mean deviation: 8.86 ms
- Median deviation: 9.21 ms
- Maximum deviation: 17.52 ms

Table 5: Statistics for the SLUE-VoxPopuli NEL dataset. The training set was constructed by augmenting the SLUE-VoxPopuli NER Shon et al. (2022) set with word-level timestamps using Montreal Forced Aligner (MFA) McAuliffe et al. (2017).

| Split | Utterances | Duration (h) | # Entity Phrases | # Utterances w/ Entities |
|-------|-----------|--------------|------------------|--------------------------|
| Train | 4924 | 14.15 | 9082 | 2844 |
| Val | 1750 | 4.97 | 1857 | 943 |
| Test | 1838 | 4.90 | 1986 | 1032 |

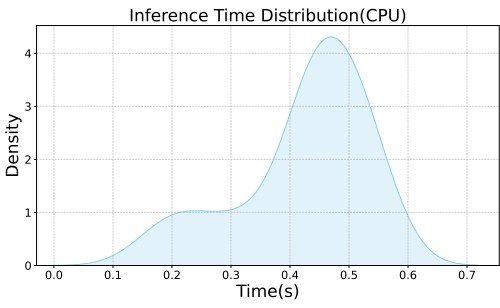

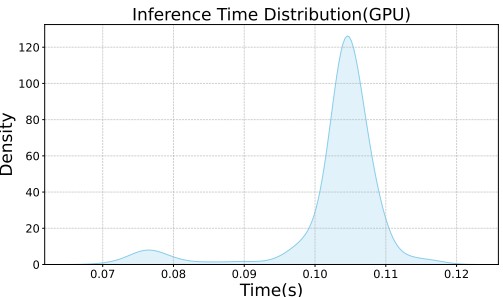

Figure 7: Inference time distribution on CPU   Figure 8: Inference time distribution on GPU

- Minimum deviation: 0.64 ms

Since DEnSNEL operates at a 20 ms frame size, these deviations correspond to less than a single frame deviation. This demonstrates that most MFA alignment errors fall well within a single frame and are unlikely to systematically cause boundary shifts larger than the model's temporal resolution.

## A.4 INFERENCE TIME

Figures 7 and 8 illustrate the inference-time distributions for DEnSNEL on the SLUE VoxPopuli NEL test set (Shon et al., 2023), evaluated on CPU and GPU respectively. We conducted the experiments on a machine with an AMD Ryzen 7 5800X CPU, featuring 8 cores and 16 threads, and a NVIDIA RTX 4000 Ada Generation GPU with 20 GB VRAM. The results reveal that GPU-based inference is faster and more consistent, with most inference times tightly clustered around 0.1 seconds. In contrast, CPU inference times are more variable and generally higher, peaking around 0.55 seconds, which highlights the advantage of GPU acceleration for near real-time applications. Figure 9 shows the distribution of audio lengths, indicating that most samples are under 10 seconds, with a peak around 6–7 seconds and a long tail extending up to 35 seconds. This variability in audio duration partially explains the broader distribution of CPU inference times. Together, these plots demonstrate that while DEnSNEL is capable of handling a wide range of input lengths, GPU deployment is more suitable for maintaining low-latency performance in practical, near real-time scenarios. Specifically, inference on GPU consistently completes within approximately 100 milliseconds per sample, enabling near real-time entity localization suitable for streaming and interactive applications. A summary of the model's complexity and efficiency metrics is provided in Table 6.

| Metric | DEnSNEL |
|--------|---------|
| Parameter count | 87.48M |
| FLOPs count | 346.2 GFLOPs |
| Real-time factor (CPU) | ∼0.083 |
| Real-time factor (GPU) | ∼0.017 |

Table 6: Parameter count, FLOPS count, latency  real time factor of DEnSNEL.

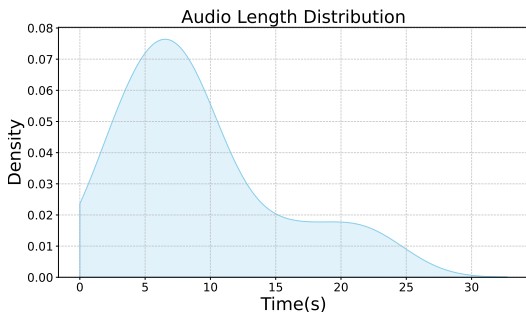

Figure 9: Audio length distribution of SLUE Voxpopuli NEL (Shon et al., 2023) test dataset

Table 7: Recall scores by named entity type on the evaluation set

| NER Type | TP | FN | Recall |
|---|---|---|---|
| GPE | 459 | 80 | 0.8516 |
| LOC | 144 | 22 | 0.8675 |
| CARDINAL | 117 | 49 | 0.7048 |
| MONEY | 4 | 4 | 0.5000 |
| ORDINAL | 52 | 4 | 0.9286 |
| PERCENT | 2 | 0 | 1.0000 |
| QUANTITY | 7 | 1 | 0.8750 |
| ORG | 182 | 88 | 0.6741 |
| DATE | 145 | 26 | 0.8480 |
| TIME | 6 | 1 | 0.8571 |
| NORP | 278 | 66 | 0.8081 |
| PERSON | 61 | 19 | 0.7625 |
| LAW | 38 | 56 | 0.4043 |
| EVENT | 14 | 5 | 0.7368 |
| LANGUAGE | 10 | 1 | 0.9091 |
| PRODUCT | 1 | 5 | 0.1667 |
| WORK_OF_ART | 3 | 0 | 1.0000 |
| FAC | 1 | 3 | 0.2500 |

A.5    PERFORMANCE EVALUATION BY ENTITY TYPE

While DEnSNEL does not explicitly predict named entity types, focusing instead on binary frame-level classification (entity vs. non-entity), we analyze its effectiveness across different entity categories using the ground-truth annotations from the SLUE-VoxPopuli NEL test set. Specifically, we compute recall per entity type by matching predicted spans to ground-truth spans, without requiring correct entity type classification in our predictions. We present the recall scores across entity types in Table 7.

The results show a clear disparity in recall across entity types: the model achieves near-perfect recall for PERCENT and WORK_OF_ART (1.00), and strong performance ($\geq 0.85$) on ORDINAL, LOC, QUANTITY, GPE, DATE, and TIME, reflecting reliable detection of short or formulaic expressions. We observe a moderate recall for NORP (0.81), PERSON (0.76), EVENT (0.74), and CARDINAL (0.70). In contrast, categories characterized by low frequency and variable phrasing, PRODUCT (0.17), FAC (0.25), MONEY (0.50), and LAW (0.40), exhibit markedly lower recall, highlighting the model's relative difficulty with infrequent, multi-word, or acoustically diverse entity mentions.

## A.6    EXTENDED ANALYSIS OF RESULTS

### A.6.1    DURATION

The input to DEnSNEL is a 30-second audio segment. As shown in Figure 11, DEnSNEL achieves consistent and reliable performance not only on audio clips of approximately 30 seconds but also generalizes effectively to shorter segments, indicating robustness to variation in input duration. Additionally, the model is not constrained by the length of the NER segments themselves. It is capable of accurately detecting named entities regardless of whether they span a short phrase or a longer phrase of the audio. This flexibility allows the system to handle a wide range of real-world scenarios, from brief mentions to lengthy, context-rich entities, without requiring segment-specific tuning (see Figure 12).

### A.6.2    HUMAN ANNOTATED TIMESTAMPS

In the Figure 13, the first row presents the human-annotated named entity labels, the second row shows the ground truth labels provided in the test set of Shon et al. (2023), and the third row depicts the model predictions. Annotating named entities is inherently challenging due to the difficulty in precisely defining their temporal boundaries. As a result, the evaluation results reported in Table 1 are influenced by the subjectivity and accuracy of the provided annotations. These observations suggest that predicting strict boundaries is inherently relative.

### A.6.3    ABLATION FOR PHONETIC FEATURE INTEGRATION

To evaluate the contribution of the phonetic feature integration, we compare model performance with and without learnable complex filter banks. As shown in Table 8, the addition of the phonetic stream yields a slight improvement in frame-level F1 score. However, this numerical gain is modest because the boundary level corrections made by the phonetic features affect only a small subset of frames (e.g., 1 to 5 frames out of a total of 1500), and thus their effect is diluted in aggregate metrics.

To better capture the value of phonetic feature integration, we present qualitative examples in Figure 10. These illustrate how the phonetic stream improves the temporal alignment of entity boundaries, especially for short spans and tightly clustered entities. While small in number, such corrections are critical for privacy preserving applications. This aligns with the design motivation of incorporating low level acoustic cues to refine entity span localization.

Table 8: **Quantitative ablation of phonetic feature integration:** While the improvement in frame-level F1 score is small, this is expected since only a few boundary frames are affected. The phonetic stream primarily contributes to more precise boundary alignment, which is better captured in qualitative analysis.

| Model Variant | Frame-F1 Score |
|---|---|
| No phonetic stream | 0.7667 |
| DEnSNEL(With phonetic stream) | 0.7674 |

## A.7    QUALITATIVE RESULTS

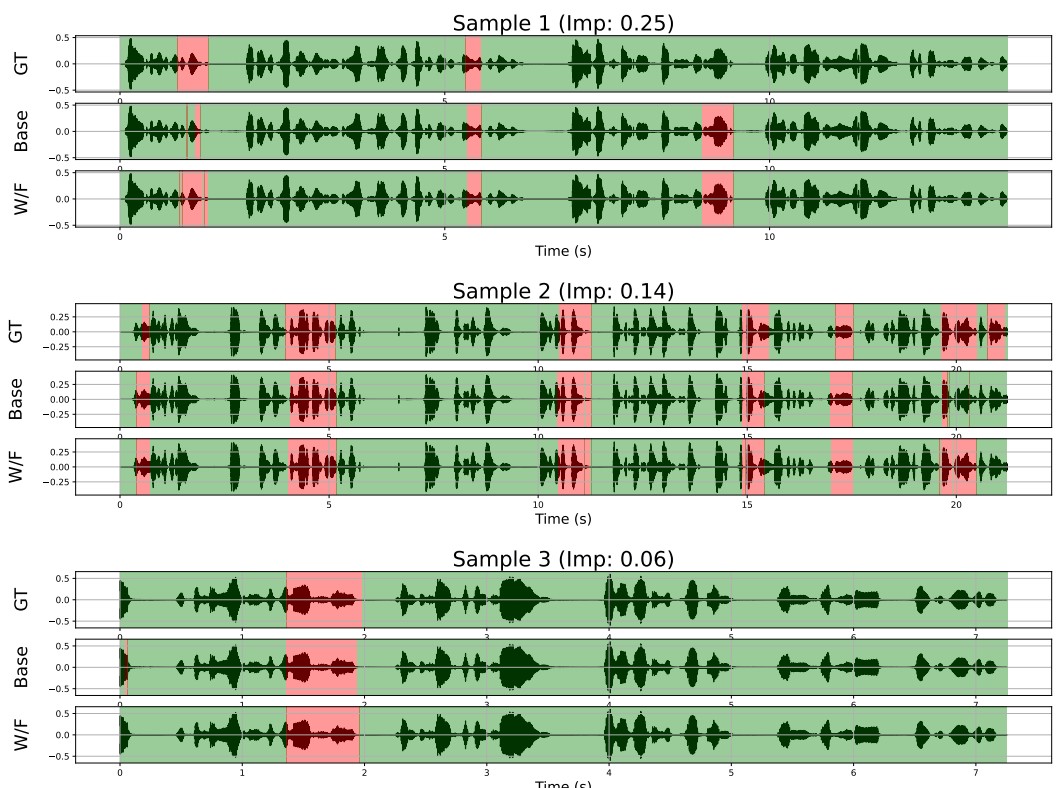

Figure 10: **Qualitative ablation for phonetic feature integration:** Red regions indicate predicted entity spans. Each row corresponds to the Ground Truth (GT), the model variant without phonetic filters, and DEnSNEL(with phonetic stream). These examples illustrate that phonetic feature integration improves alignment precision, even when such gains may not be strongly reflected in global F1 metrics.



Figure 11: Model performance on audio inputs of approximately 25 seconds (left) and 2.5 seconds (right), highlighting robustness to variations in input duration. Red regions indicate the presence of named entities in both ground truth (Label) and prediction (Pred), while green regions denote non-entity spans.

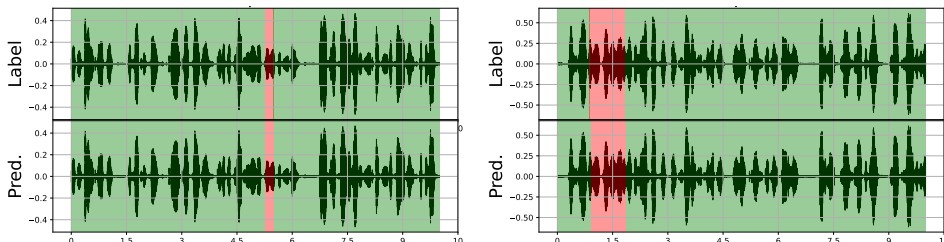

Figure 12: DEnSNEL is capable of accurately identifying named entities ranging from brief phrases to longer, contextually rich expressions. Both the left and right examples correspond to audio segments of approximately 10 seconds, with the left containing a shorter NER phrase and the right featuring a longer NER phrase. Red regions indicate the presence of named entities in both ground truth (Label) and prediction (Pred), while green regions denote non-entity spans.

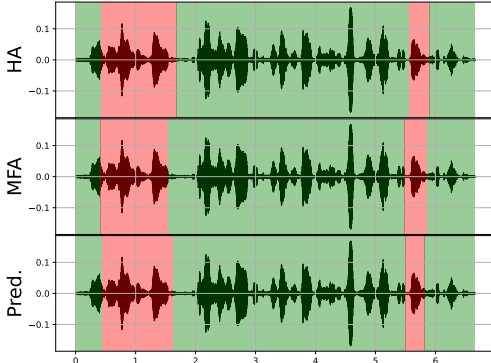

Figure 13: The top row displays the human-annotated entity timestamps (annotated by an author), the middle row shows the ground truth labels provided in Shon et al. (2023), generated using the Montreal Forced Aligner (McAuliffe et al., 2017), and the bottom row represents the model's predictions. Red regions indicate the presence of named entities in hand annotated (HA) ground truth (MFA) and prediction (Pred), while green regions denote non-entity spans.

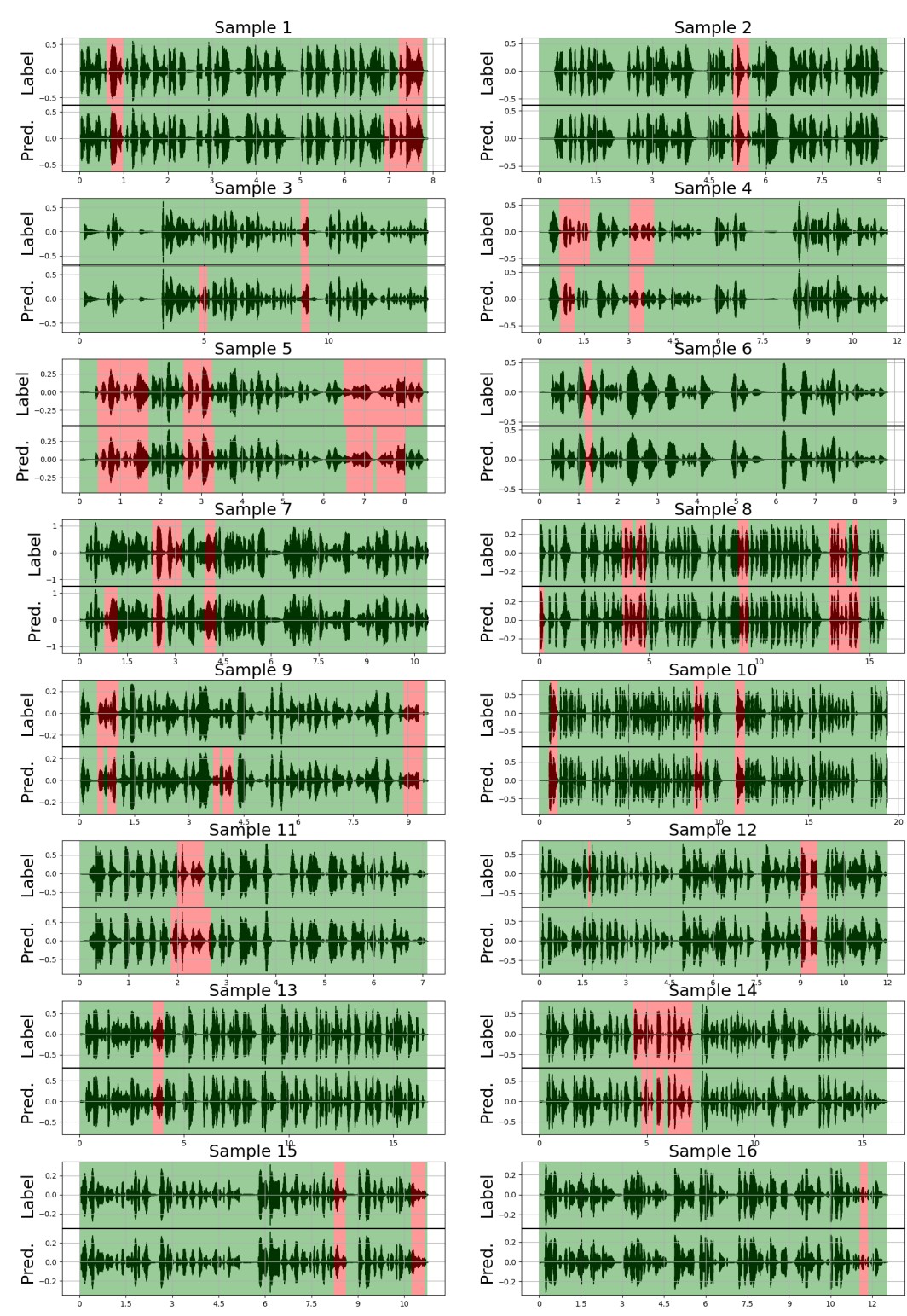

Figure 14: **Qualitative Evaluation:** DEnSNEL accurately predicts most ground-truth entity spans, with high overlap between predicted and true regions. While occasional false positives occur, the predicted spans consistently cover the ground-truth spans to a significant extent, demonstrating strong temporal alignment and high recall.

