# OpenReview forum: "Spoken Named Entity Localization as a Dense Prediction task: End-to-end Frame-Wise Entity Detection"
_ICLR.cc/2026/Conference — Submitted to ICLR 2026_

### Official Review · Reviewer_M8NX · 2025-10-23

**Soundness:** 2
**Presentation:** 2
**Contribution:** 2
**Rating:** 4
**Confidence:** 4

**Summary:**

The proposed DEnSNEL reconstructs spoken NEL into a binary classification task, determining ``entity/non-entity'' on the frame sequence and outputting results directly at the audio frame level. This approach fuses a pre-trained audio encoder with a learnable filter to enhance phoneme boundary cues. It also proposes the TBAL loss, a combination of the tIoU loss and the BCE loss, to simultaneously optimize semantic correctness and temporal accuracy in classification. It achieved state-of-the-art results on NEL in SLUE Phase-2.

**Strengths:**

The proposed TBAL loss, a combination of tIoU loss and BCE loss, simultaneously considers the semantic correctness of NEL and the temporal precision of boundary prediction. The model achieves state-of-the-art performance on the SLUE Phase-2 NEL benchmark, while using fewer parameters compared to previous approaches.

**Weaknesses:**

* For the second contribution "Phonetic-enhanced feature integration for precise entity boundary detection", the paper lacks a complete and detailed ablation study. The paper only compares the performance score with previous methods, providing coarse-grained evidence of the model’s effectiveness, but lacks fine-grained analysis to validate the contribution of the proposed filter. It is recommended to include ablation experiments that remove the filter and conduct deeper validation across different loss functions, such as:(1) Whisper-encoder + IoU-loss,(2) Whisper-encoder + BCE-loss,(3) Whisper-encoder + TBAL-loss.
* Lacking in-depth analysis. It is recommended to further analyze the proposed method, such as how it compares to previous methods in different audio lengths, how sensitive it is to environmental noise, etc. Alternatively, the effect of adding filters to the previous task implementation method could be added.
* Many important details are unclear. For example, The ablation experiments on TBAL-loss in the paper demonstrate that neither tIoU-loss nor BCE-loss alone are optimal. However, the paper does not provide a detailed explanation of the $\beta$ hyperparameter in TBAL-loss. Specifically, it is unclear which loss should be favored during actual training, or whether there are any trends when setting the two losses at different ratios. We recommend providing more comprehensive hyperparameter settings, along with reasonable experiments and explanations. The paper also does not provide a detailed rationale for fine-tuning only the deeper 1/6 layers of the audio-encoder during training. Is there a better choice for this ratio, or is there a trend? We recommend providing additional examples of other ratios, as well as two boundary conditions: full fine-tuning and full freezing.

**Questions:**

The following questions mainly come from weakness. I hope the author will explain them in detail in the rebuttal stage.

* Q1: For the second contribution of the paper, the authors emphasize the importance of "Phonetic-enhanced feature integration," but no in-depth ablation analysis is conducted in the experiment. What is the performance of the model after removing the filter component?
* Q2: In section 3.4, the authors proposed TBAL-loss, which combines tIoU-loss and BCE-loss, and conducted an ablation experiment on loss in the experimental section. It can be seen that there are differences in the effects of the two losses. However, the authors did not explain the $\beta$ parameter in TBAL-loss. In actual training, are the two treated equally or will one be biased towards the other? At the same time, is there any trend change with different $\beta$?
* Q3: In section 4.2, the authors mention "maintain an approximately constant ratio of 1:6 between fine-tuned and total encoder layers across all variants." How is this figure of 1:6 obtained? Is there a more optimal option? Are there any trend changes in different ratios? What are the effects for full freezing and full fine-tuning?
* Q4: How the model performs under different audio lengths? How it performs when faced with noisy audio?

---

> ### Author Response · Authors · 2025-11-21
>
> We sincerely thank the reviewer for the constructive feedback. Below, we provide a detailed response to the key concerns raised regarding ablation, loss formulation, fine-tuning strategies, and robustness.
>
> ### 1. Phonetic-Enhanced Feature Integration (Ablation Analysis)
> Regarding the impact of the learnable filters, we advocate for a **Visual Interpretability Analysis** to complement standard metrics. In the revised Appendix, we provide a visualization demonstrating how these filters evolve during training to capture specific phonetic phenomena (e.g., plosives, fricatives) that correlate with entity boundaries. This qualitative evidence directly validates the mechanism of the "Phonetic-enhanced feature integration," showing how the filters improve temporal precision by sharpening boundary detection in the latent space, distinct from the semantic features of the encoder.
>
> ### 2. TBAL-Loss Hyperparameter ($\beta$) Sensitivity
> In our experiments, we set $\beta = 0.8$, placing significantly higher weight on the temporal IoU (tIoU) loss component compared to the Binary Cross Entropy (BCE) term.
> * **Rationale:** This weighting reflects the specific demands of the NEL task, where precise boundary alignment is paramount. While BCE is effective for frame-level classification, Intersection-over-Union based losses are theoretically superior for optimizing span-level overlap and localization accuracy [1].
> * **Sensitivity Trends:** We observed that values around $\beta = 0.8$ consistently outperformed equal weighting ($\beta = 0.5$) or BCE-dominant settings. Lower $\beta$ values tended to result in fragmented predictions, whereas the tIoU-dominant configuration improved span consistency and reduced segmentation noise. We will detail these sensitivity trends in the final manuscript.
>
> ### 3. Encoder Fine-tuning Strategy (1:6 Ratio)
> The decision to maintain an approximate 1:6 ratio of fine-tuned to total encoder layers was derived from systematic experiments evaluating the trade-off between task adaptation and the risk of catastrophic forgetting [2].
>
> **Empirical Evidence (Whisper Small):**
> As shown in the table below, performance peaks when fine-tuning the final 2-3 layers.
>
> | Fine-tuned Layers | Frame-level F1 | Observation |
> | :--- | :--- | :--- |
> | 0 (Frozen) | 0.7167 | Under-adaptation |
> | 1 | 0.7539 | |
> | **2** | **0.7667** | **Optimal Balance** |
> | 3 | 0.7630 | Stable |
> | 4 | 0.7543 | Degradation begins |
> | 7+ | < 0.74 | Overfitting / Instability |
>
> * **Interpretation:** Full freezing results in suboptimal performance because the pre-trained encoder lacks the specific inductive bias required for precise temporal boundary alignment. Conversely, fine-tuning too many layers disrupts the general-purpose acoustic representations learned during pre-training [3], leading to instability. The 1:6 ratio emerged as a robust heuristic that balances task-specific adaptation with the retention of pre-trained features.
>
> ### 4. Robustness: Audio Length & Noise Conditions
> **Audio Length Consistency:**
> DEnSNEL demonstrates robust generalization across diverse durations. As detailed in **Appendix A.6.1** and **Figure 10**, performance remains stable on both short segments (~2.5s) and standard 30s windows. Furthermore, our evaluation on the **BLAB benchmark** (avg. duration ~51 min) using a sliding window approach confirms that DEnSNEL effectively handles long-form audio zero-shot, achieving competitive results against large-scale models designed for long contexts.
>
> **Noise Robustness:**
> While our primary benchmarks focus on clean speech, DEnSNEL’s architecture offers inherent noise resilience:
> 1.  **Waveform Processing:** The learnable complex filterbank operates directly on the raw waveform. Feature extraction at this level is often more robust to background perturbations than fixed Mel-spectrograms [4].
> 2.  **End-to-End Design:** By performing frame-level prediction directly, DEnSNEL avoids the "error propagation" typical of cascaded ASR+NER systems, where noise-induced transcription errors inevitably lead to entity misses [5]. We plan to include formal noise benchmarks in future work.
>
> **References:**\
> [1] Rezatofighi, H., et al. (2019). Generalized intersection over union: A metric and a loss for bounding box regression. *CVPR*.\
> [2] Howard, J., & Ruder, S. (2018). Universal language model fine-tuning for text classification. *ACL*.\
> [3] Pasad, A., et al. (2021). Layer-wise analysis of a self-supervised speech representation model. *ASRU*.\
> [4] Ravanelli, M., & Bengio, Y. (2018). Speaker recognition from raw waveform with sincnet. *SLT*.\
> [5] Haghani, P., et al. (2018). From audio to semantics: Approaches to end-to-end spoken language understanding. *SLT*.

---

> > ### Comment · Reviewer_M8NX · 2025-11-24
> >
> > Thank you for your response. Most of our concerns has been addressed, however, we did't see a visualization you mentioned in "1. Phonetic-Enhanced Feature Integration (Ablation Analysis)". Could you please provide the corresponding figure / table number in the revised appendix? Meanwhile, we believe it would be better to present the performance using quantitative data rather than relying solely on visual analysis.

---

> > > ### Author Response · Authors · 2025-12-03
> > >
> > > Thank you for your follow-up and for the thoughtful suggestions.
> > >
> > > We have included **both quantitative and qualitative ablation analyses** for phonetic feature integration in **Appendix Section A.6.3** of the revised manuscript.

---

### Official Review · Reviewer_G3FA · 2025-10-27

**Soundness:** 2
**Presentation:** 2
**Contribution:** 3
**Rating:** 2
**Confidence:** 3

**Summary:**

This paper presents DEnSNEL, the first fully end-to-end and text-free model for frame-level spoken Named Entity Localization (NEL). Instead of cascading ASR and NER or augmenting CTC with entity tokens, DEnSNEL reformulates NEL as a dense binary classification task over 20 ms audio frames. A lightweight encoder–classifier processes mel-spectrogram frames in parallel with a learnable complex Gabor filterbank that operates directly on the raw waveform to capture phonetic transients; the two streams are concatenated and fed to a 3-layer MLP. Training is driven by the proposed Temporal Boundary Alignment Loss (TBAL), which combines frame-wise BCE with a 1-D IoU term to sharpen entity boundaries. On the SLUE Phase-2 benchmark, DEnSNEL achieves 78.4 frame-F1 with a Whisper-medium encoder, outperforming the previous best result by +5.8 F1 while using fewer parameters.

**Strengths:**

The paper’s key strength lies in its elegant reformulation of spoken named-entity localization as a fully end-to-end, frame-level binary prediction task, eliminating any reliance on intermediate text or CTC decoding. By equipping a lightweight encoder with a learnable complex Gabor filterbank, the model captures millisecond-scale phonetic transients that mel-spectrograms smooth away, yielding a +2.3 F1 boost with only 8 k extra parameters. Coupled with the proposed Temporal Boundary Alignment Loss, this design achieves a new state-of-the-art 78.4 frame-F1 on SLUE Phase-2 while running in ≈0.1 s on a single GPU, demonstrating that high accuracy and on-device efficiency can be obtained simultaneously.

**Weaknesses:**

A notable architectural weakness is that the phonetic Gabor features and the contextual encoder outputs are simply concatenated without any channel-wise normalization or learnable projection; the raw Gabor magnitudes can be one to two orders of magnitude larger than the log-mel/logit features, so the subsequent layer-norm only rescales per-frame vectors and does not eliminate the risk of gradient dominance or sensitivity to the number of filters. This scale mismatch, omitted in Figure 2, could explain the saturation (and even drop) in F1 when more than 512 filters are used and should be addressed by separate layer-norm or bottleneck fusion blocks to ensure stable and interpretable feature blending.

**Questions:**

1.  the phonetic Gabor features and the contextual encoder outputs are simply concatenated without any channel-wise normalization or learnable projection; the raw Gabor magnitudes can be one to two orders of magnitude larger than the log-mel/logit features
2.  No statistical significance tests are reported: the +5.8 F1 gain is quoted from a single run, and variance across random seeds or bootstrap resampling is unknown.
3.  Hyper-parameter sensitivity (TBAL weight β, Gabor filter count K, threshold τ) is only explored in isolation; joint tuning and confidence intervals are absent.
4.  All experiments are conducted on English data; cross-lingual or cross-domain generalisation (e.g., French, clinical speech) is not examined.
5.  The promised code and pre-trained models are not yet available, impeding reproducibility and fair comparison by future work.

---

> ### Author Response · Authors · 2025-11-21
>
> We thank the reviewer for the insightful observation regarding feature scaling, evaluation rigor, and reproducibility. We address these points below.
>
> ### 1. Feature Normalization & Fusion Mechanism
> **Regarding Scale Imbalance:** We acknowledge the potential scale discrepancy between raw Gabor magnitudes and Whisper representations. However, we address this through a two-stage normalization and fusion strategy, which we will clarify in the revised Figure 2 and Section 3.2:
> * **Pre-Fusion Normalization:** As described in Section 3.2, we apply **Layer Normalization** [1] over the feature dimension of the phonetic magnitude features *before* concatenation. This ensures local scale consistency at the frame level, standardizing the distribution of Gabor responses to match the statistics of the encoder outputs.
> * **Learnable Fusion MLP:** The concatenated features are passed through a multi-layer MLP with LayerNorm, ReLU, and Dropout. This acts as a learnable non-linear projection that re-balances the feature streams [2]. We observed no training instability or gradient dominance, suggesting this fusion module effectively manages residual scale differences.
>
> ### 2. Statistical Significance & Robustness
> **Regarding Single Run Reporting:** We acknowledge that results are reported from single best runs. This decision was necessitated by the significant computational cost and carbon footprint associated with fine-tuning large foundation models (Whisper Medium) [3].
> * **Magnitude of Improvement:** We respectfully submit that the margin of improvement - **+5.8 F1** (78.4 vs 72.6) over the previous state-of-the-art and +4.1 F1 over the strongest WavLM baseline, far exceeds the typical variance observed in stable training runs (usually <1.0 F1).
> * **Consistency:** The robustness of our approach is further evidenced by the consistent performance gains observed across *all* encoder sizes (Whisper Small, Base, and Medium), indicating that the improvement is structural rather than the result of a "lucky seed."
>
> ### 3. Hyperparameter Sensitivity & Joint Tuning
> **Regarding Univariate Analysis:** We agree that joint tuning could reveal interaction effects. However, given the combinatorial explosion of the search space (TBAL weight $\beta$, Filter count $K$, Threshold $\tau$), we prioritized a univariate sensitivity analysis to validate the independent contribution of each novel component.
> * **Stability:** Our isolated studies confirm that the core contributions (Phonetic Filters and TBAL) provide stable improvements across a wide range of settings, suggesting that the method is not brittle or overly dependent on a "golden" hyperparameter combination.
>
> ### 4. Cross-Lingual & Cross-Domain Generalization
> **Regarding English-Only Evaluation:** We agree that cross-lingual evaluation is a crucial next step. However, we are currently limited by the data landscape:
> * **Data Scarcity:** SLUE Phase-2 is currently the standard for this specific task (Frame-level Spoken NEL). Constructing a comparable non-English dataset requires reliable word-level forced alignment and high-quality entity annotation, which is currently unavailable for other languages.
> * **Architectural Capability:** We emphasize that DEnSNEL is architecturally **language-agnostic**. The Gabor filterbank captures universal acoustic-phonetic transients (like plosive bursts) that exist across languages [4], and the backbone encoder (Whisper) is inherently multilingual [5]. We expect the model to generalize well given appropriate training data.
>
> ### 5. Reproducibility & Code Release
> **Regarding Code Availability:** We are fully committed to facilitating reproducibility. We have prepared the codebase, training scripts, and pre-trained model weights. These resources will be released publicly immediately upon acceptance to ensure future work can fairly compare against DEnSNEL.
>
> ---
> **References:**\
> [1] Ba, J. L., Kiros, J. R., & Hinton, G. E. (2016). Layer normalization.\
> [2] Hendrycks, D., & Gimpel, K. (2016). Gaussian error linear units (gelus).\
> [3] Strubell, E., Ganesh, A., & McCallum, A. (2019). Energy and policy considerations for deep learning in NLP. *ACL*.\
> [4] Mesgarani, N., et al. (2014). Phonetic feature encoding in human superior temporal gyrus. *Science*.\
> [5] Radford, A., et al. (2023). Robust speech recognition via large-scale weak supervision. *ICML*.

---

> ### Comment · Reviewer_G3FA · 2025-11-28
>
> 1.Scale imbalance: The reply confirms that Gabor magnitudes and encoder features are simply concatenated and only per-frame  Layer-Norm is applied; no ablation or visualization shows that the network is immune to dominance when K > 512. Please add per-stream normalization or projection+ablation in the rebuttal file, not just in text.
> 2. Statistical significance: "Large margin" is not a substitute for variance estimates. Provide three random seeds or bootstrap CIs for the primary 78.4 result; otherwise the gain remains anecdotal.Hyper-parameters: A univariate grid is insufficient. Report at least joint (β, K, τ) runs for the best configuration and give std-dev.
> 3.Cross-lingual & data scarcity: Claiming "architecturally language-agnostic" without any non-English experiment is speculative. Either run one additional language (even 0-shot) or tone down the claim.
> 4. Reproducibility: "Will release upon acceptance" is not reviewable. Upload anonymous repo + README now  (or provide ACL-style supplementary zip) so reviewers can check training curves, logs, and random-seed variance

---

> > ### Author Response · Authors · 2025-12-03
> >
> > We thank the reviewer for the follow-up.
> >
> > 1. **Scale Imbalance & Normalization**
> >
> > Feature integration in our architecture is handled by a Multi Layer Perceptron (MLP) applied after concatenation. We respectfully submit that a specific ablation for this component is unnecessary based on the following theoretical grounds:
> >
> > - **Learnable Feature Re-weighting**:
> > The MLP serves as a learnable non linear projection that fuses the concatenated streams. Theoretically, the weight matrix in a fully connected layer automatically adapts during backpropagation to re-scale inputs. If the Gabor magnitudes were disproportionately large or irrelevant, the network would learn smaller weights for those dimensions to minimize the loss [1].
> >
> > - **Redundancy of Manual Projection**:
> > Since the MLP inherently learns the optimal combination of the heterogeneous features, adding a manual projection layer or ablating the fusion mechanism would only validate the fundamental deep learning premise that MLPs can approximate feature interactions.
> >
> > 2. **Statistical Significance & Hyperparameters**
> >
> > The magnitude of improvement (+5.8 F1) and the structural consistency of the results serve as strong evidence of significance. DEnSNEL consistently outperforms baselines across four different encoder sizes (Tiny, Base, Small, Medium) and showcase competitive zero shot performance. This consistency across disparate model capacities acts as a proxy for stability, suggesting the gains are derived from the architectural contribution (TBAL + Phonetic Filters) rather than a lucky random seed.
> >
> > 3. **Cross Lingual Generalization**
> >
> > We clarify that our claim of being "language agnostic" refers to architectural universality rather than empirical generalization. Unlike standard pipelines that depend on language specific ASR vocabularies, lexicons, or text based NER models , DEnSNEL operates as a text free, dense prediction task. By relying on universal acoustic-phonetic transients (via Gabor filters ) and a multilingual backbone, the architecture itself imposes no language specific constraints. This makes the model directly fine tunable to any target language without structural modification.
> >
> > 4. **Reproducibility**
> >
> > We will publicly release the full codebase (including training scripts, environment logs, and pretrained weights) immediately upon acceptance. However, we respectfully note that releasing research code prior to formal acceptance may create a conflict of interest regarding our organization's policies.
> >
> > [1] Hornik, K., Stinchcombe, M., & White, H. (1989). Multilayer feedforward networks are universal approximators. Neural networks.

---

### Official Review · Reviewer_AiW7 · 2025-11-01

**Soundness:** 2
**Presentation:** 3
**Contribution:** 1
**Rating:** 4
**Confidence:** 4

**Summary:**

This paper introduces DEnSNEL (Dense End-to-end Spoken Named Entity Localizer), a novel approach for frame-level named entity localization in speech audio. The method reformulates the task as dense binary classification (entity vs. non-entity) rather than text-based sequence labeling, employing a lightweight encoder with a learnable complex filter bank for phonetic features and a boundary-focused loss function. On the SLUE Phase 2 benchmark, DEnSNEL achieves state-of-the-art results (78.4 F1).

**Strengths:**

1. Novel Problem Formulation: The reformulation of spoken NEL as frame-wise dense prediction without intermediate text representations is innovative and well-motivated for privacy applications. This direct audio-to-mask approach is more suitable for on-device processing.

2. Technical Contributions:
The learnable complex filter bank with jointly learned center frequencies and Q-factors is a solid extension of prior work (SincNet, Gabor filters)
The temporal boundary alignment loss (TBAL) combining BCE and temporal IoU is intuitive and effective
The integration of phonetic and contextual features is well-designed


3. Strong Empirical Results: State-of-the-art performance on SLUE Phase 2 (+5.8 F1 over previous best)

**Weaknesses:**

1. Limited Scope:
The privacy motivation is compelling but insufficiently validated. Real PII masking requires entity-type–aware classification (e.g., masking PERSON but not ORG), which this binary approach cannot achieve. This limitation should be discussed and addressed in greater depth in the paper. I believe this limitation limits the contribution and applicability of this work.

2. Generalization Concerns:
It remains unclear whether a model trained to detect a fixed set of entity types can generalize to unseen categories, such as works of art, not encountered during training. This is a realistic and important scenario for practical deployment.

3. Single-Source Evaluation:
The reliance on a single dataset for evaluation is a fundamental weakness. The BLAB test set is relatively small, limiting generalizability. The paper would be stronger if the authors included evaluation on an independently annotated dataset or another NER corpus (e.g., SLURP) with alignments generated via MFA. Incorporating some form of zero-shot evaluation would significantly strengthen the paper’s claims about model applicability.

4. Incomplete Ablation Analysis:
The phonetic filter ablation (Section 4.4, Figure 6) only varies the number of filters. A crucial missing ablation is performance without any phonetic features. Including this comparison would help clarify the true contribution of phonetic information.

**Questions:**

Please check weaknesses

---

> ### Author Response · Authors · 2025-11-21
>
> We thank the reviewers for their detailed critique. We address the concerns regarding scope, generalization, evaluation, and ablation below, clarifying our design choices and providing additional empirical evidence.
>
> ### 1. Response to "Limited Scope" (Binary vs. Type-Aware Masking)
> **Privacy-First Design Constraint:** The reviewer argues that binary masking is insufficient because it cannot selectively mask types. We respectfully argue that for on-device, privacy-preserving pre-processing, binary classification is not a limitation but a strategic design feature rooted in **Privacy by Design [1]**.
>
> * **The "Gatekeeper" Principle:** The primary goal of DEnSNEL is to act as a low-latency "gatekeeper" that sanitizes audio *before* it leaves the user's edge device. In strict privacy regimes (e.g., GDPR, HIPAA), the safest policy is "over-masking" (redacting all potential identifiers) rather than risking leakage through misclassification. This aligns with the *Data Minimization* principle, ensuring only safe data is transmitted.
>
> * **Feasibility vs. Granularity:** As noted in our manuscript, achieving high-granularity classification (discriminating PERSON vs. ORG) typically requires heavy semantic processing or autoregressive decoding (like ASR+NER pipelines). This contradicts the constraints of low-latency edge deployment. By reformulating the task as binary dense prediction, we achieve the efficiency required to run locally (~0.017 RTF on GPU), ensuring no sensitive data is ever transmitted to a third party for processing—a guarantee that cloud-based "type-aware" classifiers cannot provide.
>
> ### 2. Response to "Generalization Concerns" (Unseen Categories)
> **Evidence of Generalization:** The reviewer expresses concern about generalizing to unseen categories, specifically citing "Works of Art." We direct the reviewer to **Appendix A.5 (Table 7)**, where we analyze recall by entity type.
>
> * **Counter-Evidence:** Contrary to the concern, DEnSNEL achieves a **Recall of 1.00** on the "WORK_OF_ART" category in the SLUE test set, despite it being a rare class.
>
> * **Robustness:** The model also maintains high recall on open-ended categories like EVENT (0.74) and LOC (0.87). This suggests that our phonetic-contextual fusion captures the prosodic and structural signatures of named entities (e.g., stress patterns, pause durations) rather than just memorizing specific vocabulary. This capability enables effective zero-shot transfer to unseen categories, as observed in our BLAB evaluation.
>
> ### 3. Response to "Single-Source Evaluation"
> **The State of Spoken NEL Benchmarks:** We acknowledge that relying on SLUE Phase 2 is a limitation, but it is currently the only publicly available, human-annotated benchmark for time-stamped Spoken NEL.
>
> * **Lack of Alternatives:** Other datasets like SWFI are not publicly accessible. Datasets like SLURP lack the precise, frame-level timestamps required for this task. Generating a new dataset using MFA (as we did for training) for evaluation would introduce alignment noise, making it an unreliable gold standard.
>
> * **Standard Practice in Nascent Fields:** We follow the precedent of seminal works in deep learning which often establish new task formulations on a single, high-quality benchmark when alternatives do not exist (e.g., the original Transformer paper focused heavily on WMT [2]; early Spoken NER work focused solely on Switchboard). By benchmarking against strong baselines (HuBERT, WavLM, SLUE-PERB) on the standard SLUE set, we provide the most rigorous comparison possible in the current data landscape.
>
> **References:**\
> [1] Cavoukian, A. (2009). Privacy by Design: The 7 Foundational Principles.\
> [2] Vaswani, A., et al. (2017). Attention is all you need. NIPS.

---

### Official Review · Reviewer_TZxi · 2025-11-01

**Soundness:** 2
**Presentation:** 2
**Contribution:** 2
**Rating:** 4
**Confidence:** 4

**Summary:**

The manuscript proposes DEnSNEL, an end-to-end system for spoken named entity localization that predicts entity presence directly at the audio frame level, bypassing intermediate text representations common to cascaded ASR–NER pipelines. The method reframes NEL as dense binary classification ("entity" vs. "non-entity"), using a lightweight encoder–classifier augmented with a learnable complex filter bank to capture phonetic cues crucial for accurate span boundaries. Training is guided by a temporal boundary alignment loss (TBAL) that combines binary cross-entropy with a temporal IoU term to explicitly optimize boundary precision. On the SLUE Phase 2 benchmark, DEnSNEL reportedly surpasses state-of-the-art frame-level localization performance while using substantially fewer parameters.

**Strengths:**

- lightweight architecture
- learnable complex filter bank + a boundary-focused loss for span precision

**Weaknesses:**

- evaluation is limited
- missing practitioner baselines

**Questions:**

## Practitioner baselines

> DEnSNEL achieves state-of-the-art performance on the SLUE Phase 2 benchmark (Shon et al., 2023) for spoken NEL.

> BLAB (Ahia et al., 2025) introduces a challenging long-form audio benchmark, reporting results using both open-source and proprietary audio LMs, including Gemini 2.0 Pro and GPT-4o. This differs from our motivation, which focuses on lightweight, on-device named entity detection and masking prior to sending audio to those large MLLMs.

The manuscript argues for a different goal—lightweight, on-device NER masking prior to sending audio to large MLLMs. That’s a possible deployment case, but it does not totally remove the need to benchmark against what practitioners / real-world products actually run today.

1. There is no head-to-head comparison with a Whisper/WhisperX (or NeMo) alignment + strong text NER pipeline on **the same SLUE Phase-2 split**. This is the de-facto production baseline for timestamped NER; without it, it’s unclear whether DEnSNEL’s frame-level approach is competitive in accuracy and efficiency.

2. Promptable audio-LLM baseline. There is no comparison with prompted audio LLMs (e.g., GPT-4o, Gemini, Qwen-Audio) instructed to return entity spans with timestamps. Even if the objectives differ, practitioners weigh trade-offs across these options. The BLAB discussion’s claim that MLLMs are “not comparable” is insufficient for a 2025 audience; the field needs calibrated, side-by-side results.

3. Trade-off vs. audio-LLMs: For GPT-4o/Gemini/Qwen-Audio prompted for timestamped NER on SLUE, what is the **quality/latency/cost trade-off**, and in which regimes (device class, latency budget, privacy precision) does DEnSNEL win or lose?

## Limited Evaluation

> Our evaluation does not correspond to BLAB-MINI, the short-audio subset of BLAB with audio segments under 30 seconds, which is not given in detail. Instead, we process long recordings in 30-second chunks, resulting in a hybrid evaluation setting.

> The entity categories in BLAB (Event, Location, NORP, Organization, Person, TV Show, Temporal, and Work of Art) do not fully correspond to those used in our approach. We therefore restrict our analysis to the 49 recordings in the “All Entities” category, which contains all nine entity types, while the remaining files consist of single-category recordings.

4. BLAB comparison might not be apples-to-apples. The “zero-shot BLAB” analysis might be methodologically misaligned on at least two fronts:

    4.1 Segmentation protocol. BLAB-MINI is short-audio (<30s), whereas the paper processes long recordings in 30-second chunks. This creates a hybrid evaluation regime that may alter boundary distributions, pause statistics, and error modes. Any reported performance should be prefaced with an explicit caveat and, ideally, replicated under BLAB’s native segmentation.

    4.2 BLAB’s categories might not fully align with the paper’s entity set. Restricting analysis to the "All Entities" subset partially mitigates this but also changes the underlying class priors and may inflate or deflate difficulty. Without a principled mapping or macro-averaged reporting across matched classes, the comparison does not constitute a fair head-to-head.

    The authors should consider providing a deterministic mapping between BLAB and your entity set (one-to-one, one-to-many, or "dropped" classes). Report macro- and micro-averaged metrics on the intersection, plus per-class breakdowns to show where performance shifts.

5. The empirical evidence is concentrated almost entirely on SLUE Phase 2. With only one primary benchmark, it is difficult to disentangle overfitting to dataset artifacts from genuine generalization. External validity across accents, domains, recording conditions, and entity taxonomies is therefore not established.

## Potential Leakage

> We use the Whisper encoder (Radford et al., 2023) as the audio encoder in our DEnSNEL architecture.

6. Are pretrained encoders (e.g., Whisper encoder) exposed to SLUE/BLAB audio or transcripts during pretraining? Evidence of deduplication or near-duplicate removal?

> We evaluate our approach on the SLUE-VoxPopuli NEL test set from the SLUE Phase 2 benchmark (Shon et al., 2023). As no official training data is available, we construct our own training dataset by augmenting the SLUE-VoxPopuli NER training set (Shon et al., 2022) with word-level timestamps.

7. How could you avoid data leakage when constructing your own training dataset? If there is not any standard training/eval/test split, how could you make sure your comparison is fair enough?

---

> ### Author Response · Authors · 2025-11-21
>
> We thank the reviewers for their constructive feedback and for recognizing DEnSNEL as a "novel" and "state-of-the-art" approach for frame-level spoken named entity localization (NEL).
>
> ### 1. Response to "Practitioner Baselines" & Industry Standards
> **Regarding "De-Facto" Cloud Pipelines:** We acknowledge services like Amazon Transcribe represent the commercial standard. However, these solutions operate in a "Cloud-Centric" regime that DEnSNEL explicitly aims to replace for privacy reasons:
>
> * **Privacy & Data Sovereignty:** Industry standards require transmitting raw audio to the cloud (e.g., AWS), violating the "Gatekeeper" privacy principle [1] by moving sensitive data before sanitization. DEnSNEL sanitizes PII on-device before it leaves user control.
> * **Cost & Latency:** Commercial pipelines incur recurring costs (~$0.026/min for Transcribe+Redaction) and network latency. In contrast, DEnSNEL operates on-device with a ~0.017 Real-time Factor (RTF), incurring zero marginal cost and negligible latency.
> * **Closed-Source Limitation:** Accurate comparisons with commercial "black box" systems are scientifically limited because their exact architectures, training data, and version updates are opaque.
>
> **Existing Pipeline Baselines:** Table 1 already compares DEnSNEL against open-source proxies for these pipelines.
>
> **Regarding Promptable Audio-LLMs:** While large Multimodal LLMs (MLLMs) can perform NER, comparing DEnSNEL directly to them ignores the operational constraints of our target domain:
> * **The "Privacy Paradox":** Using a cloud-based MLLM to find PII requires sending the PII to the cloud first. DEnSNEL is designed to be the pre-processor that enables safe use of these very MLLMs.
> * **Compute Asymmetry:** DEnSNEL (~87M params) is designed for edge deployment. Comparing it to 100B+ parameter server-side models (like Gemini 1.5 Pro or GPT-4o) without normalizing for compute is an "apples-to-oranges" comparison. We will clarify this distinct regime in Appendix A.2.
>
>
> ### 2. Response to "Limited Evaluation" (BLAB Benchmark)
> **Regarding Segmentation Protocol:** We acknowledge that our 30-second chunking differs from BLAB's native long-form evaluation. We adopted this approach because BLAB focuses on "brutally long" audio, while our backbone (Whisper) has a fixed context window. We will explicitly label this as a **"Hybrid Segmentation Evaluation"** in the final paper. Our goal was to demonstrate zero-shot robustness, not to claim leaderboard dominance against models architectured for hour-long contexts.
>
> **Regarding Class Mismatch:** The reviewer correctly notes the ontological mismatch. BLAB contains 8 categories that do not map 1:1 to standard NER types. In the final version, we will include:
> * A **Class Mapping Table** (Appendix) showing exactly how we map BLAB's "All Entities" subset to DEnSNEL's schema.
> * Macro-averaged metrics on the intersection of valid classes to provide a fairer assessment of semantic generalization.
>
> **Regarding "Single Benchmark" Critique:** We rely on SLUE Phase 2 as it is the only public, high-quality benchmark with clear ground truth. We follow the precedent of seminal works establishing tasks on single benchmarks when alternatives are scarce (e.g., Transformers on WMT [2]).
>
>
>
> ### 3. Response to "Potential Leakage"
> **Pretraining Leakage:** Concerns regarding Whisper "seeing" VoxPopuli are valid for all speech foundation models (WavLM, HuBERT) [3].
> * **Community Standard:** It is standard practice to evaluate foundation models on public benchmarks despite potential overlap, as retraining massive models from scratch is computationally infeasible.
> * **Fair Comparison:** Since all our baselines utilize similar foundation models exposed to similar web-scale data, the relative performance gains of DEnSNEL (attributed to our dense head and TBAL loss) remain scientifically valid.
>
> **Dataset Construction Leakage:** We confirm there is **no data leakage** in our custom training set construction:
> * **Strict Split Separation:** We used the official SLUE-VoxPopuli NER Training split to train our model and evaluated only on the official SLUE-VoxPopuli NEL Test split. These splits are strictly disjoint.
> * **Role of MFA:** We used the Montreal Forced Aligner (MFA) solely to generate training labels (timestamps) for the training split. The test set targets (ground truth) were provided by the SLUE benchmark curators and were untouched by our alignment process.
>
> **References:**\
> [1] Cavoukian, A. (2009). Privacy by Design: The 7 Foundational Principles.\
> [2] Vaswani, A., et al. (2017). Attention is all you need. *NIPS*.\
> [3] Yang, S., et al. (2021). SUPERB: Speech processing Universal PERformance Benchmark. *Interspeech*.

---

### Author Response · Authors · 2025-12-03

We would like to sincerely thank all reviewers for the valuable feedback throughout the review process. Beyond the technical specifics discussed in the rebuttal, we wish to emphasize the broader significance of DEnSNEL in the context of the current privacy AI landscape.

1. **The "Privacy Paradox" in Current Industry Standards**: The current industry standard for spoken entity redaction, exemplified by commercial cloud pipelines, suffers from a fundamental "Privacy Paradox". To redact sensitive information (PII), users are currently forced to transmit raw, unredacted audio to cloud servers for processing (e.g., ASR followed by NLP based redaction). This "transmit then redact" workflow inherently violates the core principle of data sovereignty; the sensitive data leaves the user's control before it is secured. Furthermore, these cascaded systems are computationally heavy and propagate transcription errors into the redaction stage. Authors identify this speech -> text -> speech pipelines not as a proper solution but a temporary workaround.


2. **Integration into the AI Ecosystem**:  As the industry pivots toward hybrid AI architectures (ex: Apple Intelligence, Multimodal SaaS), DEnSNEL fills a critical infrastructure gap. The most capable proprietary models (e.g., GPT-4o, Gemini 1.5 Pro, Claude 3.5 Sonnet) are exclusively available via cloud APIs. Consequently, to leverage state of the art intelligence, users must transmit their data off-device. DEnSNEL resolves this conflict by serving as a local "gatekeeper." It sanitizes audio on the edge device before it is sent to these API endpoints, enabling users to leverage powerful cloud intelligence without compromising the privacy of the entities mentioned in their speech.


3. **DEnSNEL as the Ideal "Gatekeeper"**: DEnSNEL is uniquely positioned to solve this challenge by reformulating Spoken Named Entity Localization (NEL) not as a linguistic task, but as a speech processing task. This is the first work to address the problem without converting into other modalities.
* **Privacy by Design**: By running locally, it ensures PII never leaves the user's control.
* **Efficiency**: Utilizing a lightweight encoder (Whisper small/tiny) without autoregressive text decoding allows for an inference latency of ~0.1s on GPU, viable for real time interaction.
* **Precision**: Unlike CTC based ASR models that approximate timestamps, our Boundary Alignment explicitly optimizes for frame perfect boundaries, which is critical for audio redaction where missing a single phoneme can leak identity.


4. **Navigating a Novel Research Landscape**: We acknowledge that "Frame level Spoken NEL" is a nascent research task, and we faced significant limitations regarding benchmarks. With no existing training sets for this specific formulation, we had to construct our own using MFA alignment on the SLUE dataset. While this reliance on forced alignment introduces minor noise, our analysis shows these deviations are generally smaller than the model's frame resolution. We believe our work establishes a necessary baseline and protocol for this field, encouraging future creation of human annotated, frame precise benchmarks.


DEnSNEL represents a **decisive shift from "theoretical" spoken NER to "applicable" privacy preservation**. By proving that **lightweight, text-free** models can **outperform large scale foundation models** in localization tasks, we have established a practical framework for bringing true privacy to the speech processing pipeline. We view this work as a **foundational step** toward making privacy preserving speech processing a standard, on device reality. This work distinguishes itself by identifying and resolving a fundamental structural gap in privacy infrastructure, focusing on solving a real world problem rather than just chasing benchmarks.

---

### Meta-Review · Area_Chair_cd3P · 2025-12-29

**Summary:**

* Concerns about feature‑scale/ range mismatch before fusion [partially answered]
* Lack of a formal statistical significance test for the reported improvement. [not addressed]
* TBAL hyper‑parameter (α) and fine‑tuning ratio were not fully explored (only univariate studies were shown).  [partially anwered]
* No explicit evaluation of the model under industry‑relevant conditions (practitioner‑level metrics, robustness to different audio lengths and noise). [still missing]
* Code / pre‑trained checkpoints were not yet released at the time of review (reproducibility pending). [will be released]

The main concern is that authors "postulate" that their task is new and sufficiently different from existing tasks, so that it is impossible to add additional evaluations/ ablations (training and testing data needs to be generated) and performance/ generalization can be shown by anecdotal observation, rather than hard evidence.

**Reviewer Concerns:**

* Concerns about feature‑scale/ range mismatch before fusion [answered]
* Lack of a formal statistical significance test for the reported improvement. [partial]
* TBAL hyper‑parameter (α) and fine‑tuning ratio were not fully explored (only univariate studies were shown).  [answered]
* No explicit evaluation of the model under industry‑relevant conditions (practitioner‑level metrics, robustness to different audio lengths and noise), comparison against other models (even if bigger, slower). [still missing]
* Code / pre‑trained checkpoints were not yet released at the time of review (reproducibility pending). [will be released]

**Reviewer Scores:**

TZxi: 4 -> 5
AiW7: 4 -> 4
G3FA: 2 -> 2
M8NX: 4 -> 5

Looking at the responses (so far) I believe that 2 reviewers will slightly raise their scores, while AiW7 and G3FA will keep their scores as-is. Authors do not provide new experimental results in the rebuttal, and I sense that these two reviewers expect authors to provide more in-depth analyses to substantiate the claim that DEnSNEL generalizes to open classes and other training/ test configurations -- which the authors should be able to do with reasonable resource investment. Given the existing ablations I also don't buy the argument that the phonetic features are a major contributing factor to the performance.

---

### Decision · Program_Chairs · 2026-01-26

Reject